# Walking along the Sheeptrack . . . Rural Tourism, Ecomuseums, and Bio-Cultural Heritage

**Angelo Belliggiano** [1], **Letizia Bindi** [2] **and Corrado Ievoli** [1,*]

1    Department of Agricultural, Environmental and Food Sciences, University of Molise,
     86100 Campobasso, Italy; belliggi@unimol.it
2    Department of Social, Human and Learning Sciences, University of Molise, 86100 Campobasso, Italy;
     letizia.bindi@unimol.it
*    Correspondence: ievoli@unimol.it

**Abstract:** The paper deals with the issue of responsible and sustainable tourism starting from a series of Italian (and only partially French) cases of ecomuseums of pastoralism and transhumance as potential drivers for development and territorial regeneration, as well as for the promotion of experiential tourism with low environmental impact, capable of triggering participatory processes of inclusion and social innovation. Through the analysis of two Italian regions (Piedmont and Molise) and three cases (Ecomuseum of Pastoralism in Pontebernardo, Cuneo; Ecomuseum Itinerari Frentani, Larino; and the ongoing program of the Institutional Contract of Development in Campodipietra, Molise) the authors propose an interpretative model based on three main issues: the awareness, agenda, and action of a responsustainable tourism concept and on the three different subjects of local actors, tourists and policy-makers, obtaining as the main result the pre-eminence of intangible actions for development over environmental recovery and conservation activities.

**Keywords:** ecomuseum; responsustanaible tourism; sheep track; cultural heritage; rural diversification; local food; territorial regeneration; fragile and inner areas



## 1. Introduction

Sheep tracks (tratturi) and transhumant experiences are today considered above all as a cultural journey through which communities become fully involved in the process of heritagization and the planning of territorial development. With the notion of "heritagization", we encompass the complex set of processes and dynamics of constituting a cultural asset or a traditional practice as a legacy, creating and recreating cultural and historical meanings, sense of belonging and local or trans-local identities. This can mobilize a radical multi-disciplinary approach and activate rural–urban encounters and dialogues, making local knowledge and practices fully understandable to visitors. In this way, sustainable tourism, in addition to responding effectively to new tourist motivations, releases the economic potential underlying the increasingly widespread willingness to pay for rural services and products.

The so-called 'heritage turn' and the potential for a tourist exploitation of this transhumant landscape has met new forms of local participatory processes deeply connected with social innovation and new forms of territorial revitalization as ecomuseums.

Ecomuseums have been at the center of the socio-anthropological, as well as cultural, heritage management in the last two decades [1]. They represent the opportunity to concretely involve communities in a virtuous process of definition, interpretation, and regeneration of a territory, with all of its characteristics and local vocations [2].

Based on the Henry Rivière and Hugues de Varine's concept [3,4] of an open-air museum focused on the identity of a place, based on local participation and social innovation [5,6], and aiming to enhance the welfare and development of local communities, ecomuseums have progressively evolved in complex clusters of local socio-cultural subjects

(cultural and leisure associations, local tourist operators, different groups of interest) [7–11], "new peasantries" [12], and small food processing laboratories organized in nested markets [13–15], as well as motivated experiential tourists.

Urban consumers are increasingly interested in short supply-chains and high quality organic local foods, and in the storytelling concerning these products as embedded in the respective territories. They are fascinated as well by the possibility of being in touch with the very source of their food and commodities (where and how a particular agri-food product has been cultivated and processed; how and who has realized a specific handcraft as an added value). This can be largely supported through the ICT [16,17], e.g.., the adopting of an animal at distance and its products, as well as adopting a tree or a part of a cultivation, following at distance the life of a particular farm where our food is produced and so on.

Ecomuseums, albeit with a great differentiation between one country and another (sometimes even between a region and another), offer a model to manage and improve rural tourism, connecting all these aspects taken into consideration by the contemporary consumers and promoting a more sustainable way of interacting between local and foreign actors in a contemporary tourist scenario [18–21].

The transhumant tracks alone have a huge media and tourist visibility. They are deeply embedded in rural contexts and may constitute a model of a "modern" ecomuseum as a tool for promoting and managing rural tourism.

Moreover, ecomuseums are one of the most interesting issues of the debate on the 'heritagization' of natural and cultural goods. They solicit, in fact, several questions about methodology, and ask for multidisciplinary approaches. This can be done by several actors and agents, at different scales. Definition and valorization of a territory comes through the knowledge and vision of the places that social and human sciences, landscape planning, agrarian and rural economic knowledge, ethno-botanic and agri-food studies contribute in shaping. Having arisen around a theoretical nucleus matured essentially in the context of the museology and environmentalist reflection of the French matrix and, in particular, from the work of Hugues de Varine, Georges-Henri Rivière, and Serge Antoine in the late 1980s [22], the ecomuseums had, since their first conceptualization, an inextricable relationship with the themes of conservation and enhancement of landscapes, of tangible and intangible assets, with ecological and social sustainability, with a new and more participatory ideas of heritagization processes. At the same time, however, despite forty years of reflections and debates, the concept of ecomuseum risks remaining blurred and confused with that of ethnographic museums or ethno-anthropological interest, which in reality can often be very different and come from very little intellectual and social genesis in line with the original spirit of the ecomuseum's proposal. In fact, it is important to distinguish between small peasant and community museums scattered in various peripheral regions of Europe [23] and the territorially innovative design and participatory narration of the ecomuseums.

In Italy especially, the regulation of ecomuseums has been rooted at a regional scale since 1995 (first Regional Law in Piedmont), starting from the geographically near French examples and providing, since the first experiences, a strong interaction between civil society and institutional levels, the necessary integration between landscape and social aspects, as well as between material and immaterial heritage. From 1995 to the present, almost all regions in Italy have elaborated and signed regional laws on ecomuseums [19], some of them insisting their social and cultural value, others focusing more in their role as rural and territorial development drivers and tourist destination developers. Ecomuseums can be today a powerful tool of interpretation and "mise en forme" (formatting) for landscape and cultural, offered through a strong interactive process with local communities [24]. They activate different museological representations and trans-disciplinary competences, public/private entity interactions, and the new innovative building of a tourist destination between a local dimension and the "global hierarchies of value" [25]. At the same time

ecomuseums can also be the incubators for new and radical political approaches of cultural and social innovation in the territories [26,27].

In this sense, we explored the issue of transhumant pastoralist routes as an ecomuseum opportunity in the context of rural tourism.

## 2. Theoretical Framework

Probably the first attempt to clearly define rural tourism can be attributed to Lane [28], who defined it as a type of tourism located in rural areas by identifying some key characteristics (between countryside and small towns, small accommodation facilities scale, relations with the local population, interaction with the environment, history, culture, etc.).

In general, rural tourism is a complex phenomenon that has significantly emerged in recent decades. Broadly speaking, it concerns the movement of people to rural areas to carry out a certain number of activities. These activities (e.g., walking, horse riding, typical food preparing and/or tasting, etc.) may constitute a specific experience, depending on the characteristics of the destination in terms of environment, agriculture, food, lifestyle, cultural heritage, etc. Underlining the phenomenon, there are important customer needs. As shown below, these needs make the phenomenon quite substantial, at least they did in the Italian case, and are therefore able to generate an important economic impact. For this to happen, it is crucial that the design of the tourist product, i.e., [29–31] the combination of tangible and intangible elements (natural and cultural resources, facilities, etc.) "around a specific center of interest which represents the core of the destination marketing mix and creates an overall visitor experience including emotional aspects for the potential customers".

### 2.1. Motivations of Tourist and Rural Touristic Models

The key characteristics of rural tourism underlie different tourist approaches or motivations, which we could define as naturalistic and cultural. In fact, most of the rural tourist demand would seem attracted above all by the wealth of the natural and landscape resources of the places, which are associated with the most symbolic and/or evocative elements of the rural world (locations, menus, events and festivals, etc.,), enjoyed by the tourists with basically hedonistic and consumerist attitudes. Another part of rural tourism demand, on the other hand, albeit a minority but in strong growth, is instead driven by significant cultural motivations, consisting of the need for an in-depth knowledge of the rural contexts visited, which is also associated with a strong desire to establish an empathic relationship with residents, through their involvement in their daily life.

The orientation towards one of the two models therefore constitutes a decisive choice in defining the tourist diversification strategies of local economies oriented towards territorial regeneration [32], since the possible prevalence of the first model would not be without interference of the conservation of cultural heritage, especially the intangible type, which instead would be functional to the development, or at least to the coexistence, of the second model [33]. The latter, in fact, recognizing the intrinsic tourist potential of the presence of multiple local identities linked by practices or traditions, differs from the first model, which we could consider merely recreational, as it is not limited to the emotional or simply sensory aspects of the landscape, but turns its interest to the reasons that generated it or that perhaps transformed it over time, seeking its own itinerary through the stories and interpretations of privileged witnesses, who, in rural contexts, are mainly made up of older farmers and/or artisans [34,35], holders (often) unaware of a precious store of tacit knowledge, indispensable for the cult-rural tourist as keys to reading and interpretation of the territory.

Starting from these evidences, the most widespread analytical approach has developed by considering agricultural characteristics, small scale, cultural traditions, sustainable processes, lifestyles, differentiation of the tourist product, etc. A first point on the question was proposed by MacNulty [30] (in WTO—World Trade Organization—2004) with a

scheme that placed the rural tourist community at the center, considering four areas of offer: countryside, heritage, activities and rural life.

Ultimately, even without going so far as to say—though it would not be entirely wrong—that it is an umbrella concept [36], we can say that it is at least a very complex, quite difficult to define phenomenon, which includes "adventurous" education, sports, health, artistic and cultural aspects, etc., [30]. The definition of the WTO, that: "Rural tourism is a type of tourism activity in which the visitor's experience is related to a wide range of products generally linked to nature-based activities, agriculture, rural lifestyle/culture, angling and sightseeing", reiterates this complexity, specifying what the rurality contributes [37].

### 2.2. Rural Tourism as a Creative Sustainable and Responsible Experience of Leisure

In Europe, an integrated approach to rural tourism is affirmed in its close connections not only to natural resources but also human, social, and cultural characteristics of places and tourist destinations [38–41]. This perspective introduces the dimension of responsibility alongside that of sustainability, which, as is well known, implies the involvement of the local communities themselves as well as their respective government institutions [42]. Responsible tourism is in fact based on strategies and policies based on sustainability that should stimulate appropriate and oriented behavior, not only towards respect for environmental resources, but also towards increasing local empowerment and awareness of the possible ethical function of rural tourism.

"Sustainability is a paradigm for thinking about the future in which environmental, social and economic considerations are balanced in the pursuit of an improved quality of life" [30].

"Involvement" was recognized as an issue of the tourism product even before the use of the term "experiential" [43]. This expression was coined in the early 2000s in a business environment and then linked to Pine and Gilmore's elaboration on the experience economy [44]. This has led to the definition of standards (distinctive elements) relating to this type of tourism [45]. Consequently, in the tourism product, the 'experience phase' is clearly distinguished from that of 'choice' and the innovative tourism businesses are those based on co-production [46]. Thus, creative tourism is based above all on networks and on the value created by the match between producers and consumers [47]. Ultimately, the value of the experience includes the resources that the tourist himself, other tourists and the guest add to the process of creating the experience itself [48].

Sector players have, in fact, been increasingly identifying a growing demand for more authentic and active travel experiences, amplified by new technologies [49]. In this regard, it is noted that new ICTs enable the integration of traditional with new "voluntary" tourist information [50], such as reviews.

The rationale of experiential tourism is identified not only in the search by consumers for more authentic experiences, but also in the increase in competitiveness within the sector, aiming at growing innovation [51]. On this basis, specific tourism segments have been investigated [52].

Assuming a differentiating approach, from the consumer's point of view, the experience is personal and must remain in his long-term memory [53], as unforgettable, extraordinary, etc. On the producers' side, there is a matter of the design (creation) of the experience. The creation of value comes from the cooperation of different producers and the participation of users [54,55]. In this paper, we consider how exceptional emotions arise [56], and the creation of value as it is related to the experience of tourism [57], being in tourism different from other services. The expense, time, and effort employed are justified precisely in the pleasant moments obtained, in the experiences the consumers contribute to building a narrative [58]. The experiential environment (setting, sphere) is much more than a physical fact, but it includes consumers, producers, amenities, etc. Research must therefore identify the different roles of the different actors and resources in creating this

experience [30]. Among these, local food products [59] and the role of ICT in the creation of events [60] must obviously be considered.

This implies the creation of networks and consequent participatory issues, ensuring sustainability to the process [61] and facilitating original forms of development.

Networks, in fact, underpin new combinations of local resources (economic, socio-cultural, and environmental), engaging different community actors in the same strategy by experimenting with new forms of cooperation and collaboration, which could increase the success of the activities in which they are individually committed [62]. Among these activities, a pre-eminent role is assumed by traditional agro-food productions [63], which are recognized as having various properties capable of increasing the overall sustainability of the process [64], such as the tourist appeal of typical food and wine [65], the consequent dynamic effect on the local economy, as well as the reduction of the ecological footprint connected to the exclusive processing of local raw materials [66].

In this light, agro-food production is an important source of positive externalities that rural tourism manages to internalize more or less efficiently, due to the availability and ability of stakeholders to participate in the process [67], or from the ability of the same to identify and exploit the economies of scope underlying the use of the same territorial resources.

We increasingly observe, throughout scientific literature review and ethnographic observation, a frequent overlapping between rural activities and other new forms of tourism (such as experiential tourism, for example). Working on extensive sheep and goats as well as cow farming could therefore mean the need to deal with the depopulation of the inner and fragile areas of the Apennine ridge, fighting their increasing marginalization, a backward idea of being a shepherd who today can instead be tinged with new aspects and expectations.

At the same time, the theme of transhumant sheep farming raises opportunities for a new form of experiential and slow tourism that underlie some of the newest sensitivities and cultural needs, such as the conservation of biodiversity and the landscapes linked to these paths, as well as the artisan practices connected to this traditional practice, as well as ensuring animal welfare [68].

The working hypothesis that is emerging, therefore, is whether the revitalized transhumance can be conceived as a form of 'responsustainable' tourism and, as such, is narrated and practiced by some experiences that, in some ways, are anticipatory.

We therefore tried to verify whether the ecomuseum (of transhumance) can be considered as a container of social practices, coordinating cultural and recreational activities and proposals of territorial promotion and regeneration, aimed at facilitating and supporting forms of tourism and consumption (food, crafts), connected to the territory while having little impact, and being perfectly in line with the vocations and desires of the local population.

On another front, we tried to understand the possible criticalities and reductionisms of an operation explicitly aimed at promoting tourism along the sheep tracks, but characterized by a strong centralization and low incidence capacity of the local communities in the enhancement process.

## 3. Objectives

The paper therefore deals with factors influencing the capability of ecomuseums to be a model of development of responsible and sustainable tourism focused on transhumant pastoralism. More precisely the paper analyzes ecomuseums of pastoralism, according to Mihalic's definition of "responsustable tourism" [42,69].

In a neo-liberal context, there are some asymmetries and occasional contradictions between the economic and environmental sustainability of some uses of territories and landscapes. In sustainable tourism, the aim is to achieve a balance between socio-economic interests and ecological resources. Similarly, the purpose of responsible and sustainable tourism is to create the conditions for a harmonious coexistence between the expectations

of tourists/visitors and the desires of the territories and local communities both in terms of the conservation of their own traditions that of the aspiration towards local development and regeneration.

In this context, we tried to investigate the three main principles of responsible and sustainable tourism: awareness of sustainability and ethical principles applied to tourist destinations and proposals; the political dimension implying local communities' participation, consensus, and agency in tourist proposal implementation; responsible tourists' satisfaction [30].

Then, we tried to figure out how the routes and buildings, practice and stories, of different municipalities and rural, mountainous areas are narrated and presented through a powerful storytelling design, putting together shepherds, cheesemakers, historians and anthropologists, as well as artists and communicators.

Moreover, we analyzed how the restitution of pastoral ways of life intersects today with the local agri-food and handcrafts market, with traditional cultural expressions (chants, performances, oral poetry, etc.,) being increasingly transformed and reshaped as tourist revivals and events.

We considered transhumant ecomuseums in their different forms and variants, an extremely attractive and powerful "use of the past" [45], often reconciling local populations with a practice historically linked to backwardness and hard work, now recuperated in the sweetened and commodified form of a tourist walkways landscape [70].

Being that ecomuseums are deeply connected with local communities' involvement, we analyzed how the actors participate in the real building process of ecomuseums of transhumance and pastoralism in a cultural, as well as environmental, sense.

Implications in the consumers' choices about primary products (milk/meat/wool) and their commitment with animal welfare were considered, as well as the role of transhumance and extensive pastoralism implications in environmental safeguards, low-carbon emissions, and traditional culture conservation and enhancement. At the same time, we assisted in the final years of the growth of "return shepherds" and new pastoralists connected with environmental activism, a sense of belonging and community, food ethical production, and responsible consuming.

Moreover, our interest was to explore the relation that the transhumant ecomuseum model has with the enhancement of slow and experiential tourist models, with the ICH UNESCO list inscription, which is supposed to engender particular global visibility of transhumance all over the world, particularly through global media coverage, regional/national funds, and so on.

## 4. Materials and Methods

The choice of case studies, starting from the objectives and assumptions previously illustrated, was to verify the ecological, economic, and socio-cultural sustainability of ecomuseums connected and based on pastoralism and transhumant paths, essentially observing two territorial contexts and (within them) three specific cases.

The first territorial area taken into consideration was that located in the extreme north of the country, in the Alpine area of Piedmont (Figure 1), where we observed the performative capacity as well as the territorial and community shaping of the ecomuseum of pastoralism in Pontebernardo (EPP) in the Municipality of Pietraporzio, located in the Province of Cuneo, and characterized by historic mountain sheep tracks between the valley and the mountain pastures and an ancient horizontal transhumance—the Routo—between the mountain areas of the Alps and the plains of southern France corresponding to La Crau, presently the headquarters of the Maison de la Transhumance [71,72].

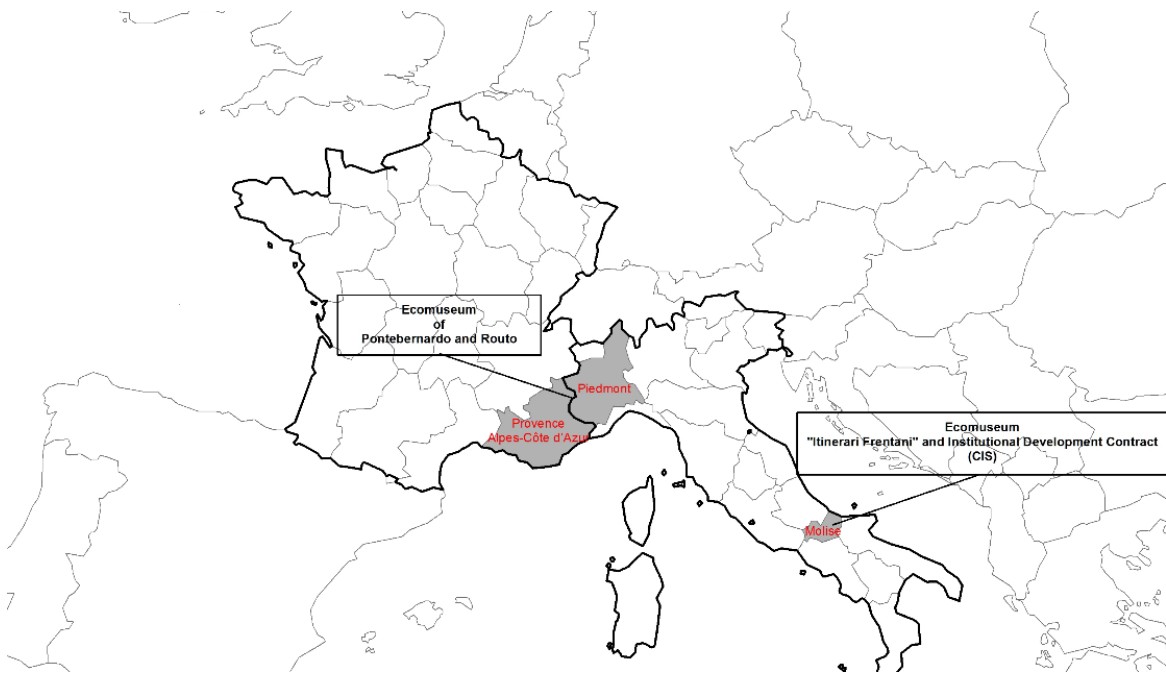

**Figure 1.** Geographical location of case study areas (map by Giuseppe Pistacchio).

The second territorial area was that of Molise, an essentially internal and mountainous region (Figure 1) located in the central–southern Italian Apennines and historically crossed by multiple sheep tracks that allowed for the movement of a huge amount of native breeding sheep, goats and cattle from the Apennine mountain areas to the Apulian plains of Foggia, where the Dogana della Mena della Pecore, an institution established during the reign of the Aragonese was once located, and which continued until the progressive decline of the practice in the second half of the nineteenth century, connected to the collapse of the value of wool and the consequent reorganization and remodeling of crops in southern Italy. In Molise we observed and monitored two experiences:

1.  That of the "Ecomuseum Itinerari Frentani" (henceforth appointed as EIF), a decennial experience of cultural animation and enhancement of local territories and communities, which in the last three years has dedicated particular attention to the recovery, rediscovery, and storytelling of the pastoral routes;

2.  The Institutional Development Contract (henceforth appointed CIS, Contratto Istituzionale di Sviluppo), an articulated national policy invested in 59 municipalities in Molise, provided with a large amount of funds directly allocated by the national government for the realization of a plan for the "Tourist development of the Molise pastoral routes". The Plan was recently developed and refined after a rather long and cumbersome phase of gestation and definition. No project for an ecomuseum has been developed up until now, despite the proposals advanced in this sense by some associations and, above all, by the University of Molise and the Interdisciplinary Research Center BIOCULT.

The two territorial contexts have been chosen based on two main reasons.

Piedmont was the first region in Italy to enact a regional law on ecomuseums in 1995 and is still the region with the largest number of officially registered ecomuseums, supported by specific regional funding and support policies. At the same time, Piedmont region and in particular the area of Cuneo, where the ecomuseum of Pastoralism is placed, has shown specific growth in the diffusion, recovery, and use of agritourism structures in the last five years that suggests a model of responsible and sustainable rural tourism [73].

Molise is one of the southern regions of Italy where sheep tracks and pastoral routes (tratturi) are more represented and well conserved, although, even in this context, they are

exposed to risks of expropriation, dispersion and abandonment, or a failure to preserve them. At the same time, the Molise region passed a regional law on ecomuseums, in line with other Italian regions, in 2008 (L.R. 28 April 2008, n.11), but the implementing decrees of this law has never developed or published, which has in fact resulted in the inconsistency of the regional policy and support action and in the impossibility for those who animate ecomuseum-type initiatives to see their work in favor of the territory recognized or financially/politically supported. Contrary to what was observed for Piedmont, in Molise, rural tourism and agri-tourism, after a diffusion and development essentially connected with the LEADER strategy, have registered a setback, and many of the structures opened at the time have seen a remodeling of their intended use and the end of the activities started, thanks to the support of the rural development strategy.

In the first phase, the empirical analysis was carried out with an ethnographic methodology and first-hand data collection through:

1.  Direct and participatory observation of the experiences put in place by the various ecomuseums studied (EPP, EIF).
2.  Direct participation to preliminary meetings for the preparation and dialogue with the national funding agencies and with the regional framework for the CIS.

In a second step, which further focused the objectives of the paper, specific information about research objectives was collected through interviews with key actors (see below).

The interviews were carried out by the authors in 2020 and the first months of 2021, explicitly returning to the topics already discussed and observed in the context of other meetings and comparisons in the field. The interviewees were involved and were made aware of the specific purpose of the interview and accepted this commitment consciously and with considerable enthusiasm.

The considerations of a more critical nature (the interpretation of the information collected) were developed during a long cooperation between the authors and a framework of numerous national and international research and action projects dedicated to sustainable territorial development.

Research methodologies therefore included:

(1) Background research and scientific literature on rural tourism and sustainability; eco-museums and their socio-cultural and economic impacts on local actors and territories; interpretative framework;
(2) Monitoring public acts launched and discussed in the various shared offices;
(3) Recording public issues and debates raised around various initiatives;
(4) Carrying out ad hoc interviews with promoters, actors, animators of the various associations, institutions, and private subjects involved in the different geographical and socio-political contexts;
(5) Analyzing and interpreting the ethnographic and socio-economic data based on three main pillars of sustainability: environment, socio-culture and economy;
(6) Analyzing three types of agents: local actors, tourists, and policy-makers;
(7) Applying to our research Mihalic's triple-A model: awareness, agenda, and action [42].

Background research, as well as interviews and ethnographic and socio-economic observations, were aimed at identifying the pivotal elements of the triple-A model (awareness, agenda and action) in the various cases, outlining the sequence of actions put in place for the building process of an eco-museum and the levels of participation involved, the diversification of its cultural proposal over time, and the more or less intense adherence to the desires of the territories and the satisfaction of tourists looking for a direct experience of contact with pastoral practice and with the forms of life connected.

## 5. Two Territories, Three Cases

### 5.1. The Ecomuseum of Pastoralism in Pontebernardo, Italy

In effect, there are already ecomuseums of transhumance and pastoralism in Italy that we can consider and evaluate as a best practice: walkways, training camps and farms, natural and ethical food, and other derived production as in the EPP [Appendix A].

Stefano Martini, during a guided tour of the EPP in 2019 and then in a virtual visit organized on the occasion of an international course organized as part of an international project of the University of Molise, coordinated by two of the authors, showing the different rooms of this territorial interpretation centre, points out "the value that the mix between biodiversity recuperation (Sambucana livestock) at the beginning and historical and anthropological knowledge has allowed to the community to find back reasons for taking care of local pastoral practice as an invitation, above all, towards young generations, hoping they'll decide to come back in small mountain villages and become shepherds". At the same time, "the very definition of "Ecomuseum of Pastoralism" has been chosen to revitalize cultural and environmental heritage and make it a pivotal resource for local tourism and sociability" (Stefano Martini, interview of 17 November 2020).

The ecomuseum (documentation, research, info-point) offers the visitor a broad image of the phenomenon of sheep breeding and sheep farming, with a substantial part related to the illustration of local reality, particularly referring to products and flavours. Here, there is also the store, recently installed, for the purchase of precious products of Sambucana sheep wool and a multifunctional room for ecomuseum activities. The creation of the ecomuseum of pastoralism began in the nineteen-eighties, thanks to the activities of the mountain community to revive the activities of traditional grazing and wandering pastures historically present in the area, and to revitalize the lines of native sheep, particularly suitable for the conformation of the territory of this community.

Among the most important activities that began thanks to the revitalization process of wandering pastures and, subsequently, to the inspiring and organizing framework of the ecomuseum, strongly supported by the innovative Piedmont regional law for the actions of the ecomuseum, these included:

- The resumption of production of the Sambucana sheep and some of the derived products in accordance with traditional and sustainable production styles (cheese, meat, meat products, and handicrafts related to wool and typical objects of pastoralist communities);
- Awareness activities, research, socialization, and dissemination of local history and traditional culture (proposal of tourist routes, educational workshops in schools, entertainment activities for the local population, a scientific magazine, brochures and illustrative guides);
- Incentives, support and productive interaction systems with craft activities related to the territory (small local businesses, cultivation of aromatic and medicinal herbs, a soap factory);
- A dairy production laboratory;
- A sheep meat processing laboratory;
- A spreaded hotel (albergo diffuso);
- A tourist proposal of trails and visits to characteristic places of the area based on the proposal of territorial knowledge developed by the heritage community, an active protagonist for the ecomuseum.

When visiting or reading the tourist brochures of the Stura Valley, one can clearly perceive the influence of the ecomuseum and the concrete participation of the local population in territorial regeneration and the productive activities present in the area.

Among the public and private, institutional and community, associative and informal actors that have contributed to the realization of the ecomuseum and the many activities that have developed from it, we find:

(a)　The ecomuseum of pastoralism of Pontebernardo (leader of the process);

(b)    The Municipality of Pietraporzio (as a smallest scale institution, which has been protagonist of some of the revitalization actions);

(c)    Consortium of the "Escaroun"—"Escaroun", in the local dialect, is the sheep that separates from the flock to go looking for the best quality grass; we could define it as the best, or a different part of the flock that separates from the rest, since it is better than others;

(d)    Mountain community of the Stura Valley (CN) (today known as Union of Mountain Municipalities: leader of some projects for the promotion and reactivation of productive activities and European financing lines, such as the recovery of the production of Sambucana sheep);

(e)    Piedmont region, advisor on the Culture/Office of the Regional Ecomuseums (promoter of the first regional law in Italy on the ecomuseums, financier of the creation of the ecomuseum and holder of the carried out activities);

(f)    University of Torino, Department of Agrarian Studies, masters in Alpine cultures (research projects and proposals for recovery of mountain activities in this area);

(g)    Relations with other structures and subjects promoting revitalization activities of the traditional pastor and transhumant, as the Maison de la Transhumance in la Crau is, for example, the geographically corresponding French side of the Routo, and the traditionally pastoral route of the transhumance of the shepherds is in this region);

(h)    Agricultural technical schools (teaching laboratories, field visits, etc.);

(i)    Local associations of culture, tourist valorization of the territory (musical groups, connoisseurs of the territory, walkers, artists, etc.).

The EPP gives us some interesting ideas to reflect on the potential related to the recovery and revitalization of rural activities that are deeply rooted in the history of the territory, which allows us to clearly identify some good practices. Through historical and anthropological investigation, we can in fact give consistency, suggestions, and content to the recovery and improvement of these activities, valorizing the interaction of local institutions and informal community groups with universities and research centres situated in the territory and particularly dedicated to this type of research object.

Meanwhile, this gives importance to the recovery of biodiversity and animal species, as well as plant products specifically linked to the history of the territory, not only because they are more suitable and adaptable to the geographical characteristics of the territory, but also because they are characterized at the territorial level and, therefore, are suitable to be promoted as typical, local products, better able to be placed differently and effectively in the supra-local markets of food and artisanal products and especially in those related to groups of joint and responsible consumers.

Urban consumers are increasingly interested in natural milk/meat/wool products, committed with animal welfare and environmental safeguarding through extensive farming practices, low-carbon emissions from extensive breeding, and with ethical concerns towards the local and cultural communities' respect and enhancement. Analogously, legal debate on common territories and common lands has been considered, as well as the consideration of the natural, human, and culturally local resources needed.

In recent years, we have assisted in the interesting changes in this area, particularly with new or 'return shepherds', the preparation of a National School for Pastoralists, in line with similar experiences already provided in France and Spain, a strong animal welfare activism concern, as well as attention to healthy/natural and even ethically produced food from the part of increasingly more informed and responsible urban consumers.

At the same time, as already stated, EPP is deeply engaged in a cooperation with the French Maison de la Transhumance, dans la region PACA, plane de la Crau. These two somehow different structures cooperate in a common project called La Routo, representing a diversification and a transformation of both the Maison's and the Ecomuseum's actions in a direction more tied to the tourist value of these activities. "La Routo is a network of pathways inspired and centred around pastoralism and wool supply chain that has allowed us to regenerate the image of the Maison as an exclusive garrison of the past, grasping the

opportunities and potential for economic revitalization of the territory in the present though the reuse of an almost obsolete material such as wool, for the creation of technical clothes for trekking. In this sense, a tourist pathway has revitalized some productive activities and enhanced the innermost and depressed areas characterized historically almost only by sheep farming . . . " (Interview of Patrick Fabre, 24 February 2020).

The Routo was initially financed by the LEADER Cooperation Program, through an integrated project by 4 LAGs, led by the LAG Terres Occitanes and by the Maison itself, in close relationship also with the directions of the regional parks of the valley. Nonetheless, la Maison de la Transhumance is more thought as an agency in the development and revitalization of pastoral activities than of tourist increase, even though the process of heritagization has engendered, for example, in the case of project La Routo, the regeneration of wool handcraft and tourist proposals for the territory [74–76].

Mauro Bernardi, head of the Unione Montana della Valle Stura, on his turn during an interview, carried out in November 2020 the framework of the International Online Course of the Erasmus + EARTH project, retracing the relevant stages of the construction of the ecomuseum and highlighting "the importance that a private entity such as the Consortium of Sambucana sheep's breeders—although solicited and stimulated by the spontaneous collective that would later give rise to the ecomuseum itself—and subsequently the importance and support of the University both for the selection and help in improving the finally, the Terre d'Oc LAG (Valle Grana, Val Maira, Valle Stura), which for a first phase supported the farmers and the consortium in this action of economic, productive and tourist regeneration, then resulting in the form of the Ecomuseum when the law of the dedicated Region allowed it (interview of Mauro Bernardi, 24 November 2020).

In the three-pillar "responsustable tourism" model, we applied to the analysis of the case studies the EPP (and the connected French experience of the Maison de la Transhumance), who have presented a case that is environmentally, economically and socio-culturally efficient. The ecomuseum, in effect, has enabled the local community to develop rural development processes that, more than a renewed autochthonous livestock production, has opened a new line of handcrafts (wool products), a growing tourist offer through recuperation of ancient buildings for agri-tourism, and a respect of environmental and historical building material constraints, in line with the recuperation of the local knowledge–practice system.

### 5.2. The Transhumance Heritage of Molise: An Opportunity for an Ecomuseum of Pastoral Pathways as a Territorial Developer

In the Italian region of Molise, transhumant tracks are recognizable even if dramatically damaged by infrastructures despite the prohibitions to build or cultivate, but only very few families still practice the transhumance [Appendix B].

It is presently interesting to evaluate how the UNESCO nomination has impacted the safeguard and valorization of transhumance and to what extent local communities on one hand, and policymakers on the other, have been really involved and counted on in this process. The very first perception is that the path of heritagization has been essentially led and realized through a very centralised and hierarchical process [77], which is, finally, somehow opposite to the original ratio of the 2003 intangible cultural heritage convention. At the same time, it is interesting how much the global nomination and recognition has determined a special visibility and really impacted on the diffusion of territorial promotion through 'slow tourism' offers along the transhumant tracks and among farms and small and medium breeding enterprises.

#### 5.2.1. The "Ecomuseum Itinerari Frentani" in Larino

In the south–central area of Apennines, and particularly in Molise, during several visits and ethnographic data-collecting periods, we observed small and bigger groups of walkers, horse-riders, and bikers involved in these pathways, and many people involved in public and ceremonial events linked to transhumance, which are still present in the territories (Carresi—oxen-charts races, holy processions, and so on). In many cases, it is a

non-religious way of walking along the transhumant paths, searching the ancient stories of the places, traditional songs and music, handcrafts, and food and conviviality. In some cases, the ecological engagement is coupled with leisure as well as the deeper knowledge of the local 'tipicality'. It is the case of EIF, an association with a strong participatory approach which has been committed for a decade in a powerful experiment of interpretation and valorization of the local environment and culture. Local communities—although largely detached today from the transhumance—still conserve many traditional practices such as food traditions and narratives (oral poetry, folk songs and dances, and so on) even despite the dramatic change, they have assisted in the last decades, and 'transhumant shepherd' is somehow still the model of the Molise rural people, though, in practice, the local population try essentially to emancipate themselves.

The model of EIF is:

(1) Deeply embedded in the local territory and community, even not officially recognized by the region Molise because of the lack of the regulatory assets of the regional law.

(2) Based on spontaneity and participation, makes strong reference to social networks as a primary form of dissemination and construction/maintenance of an emotional bond and loyalty with the ecomuseum experience [Appendix C].

(3) The ecomuseum does not receive any public financial support nor has it ever received it; it does not provide for any form of payment for the organization of visits, other than small informal support fees for companies that organize tastings, representations, process demonstrations along the way for the production of food, or other handicrafts or places of accommodation upon which you can rely on the way.

The project has seen growth in the last years, not only in the number of visits and organized itineraries with their different shades and theming, but also in the increase in companies that show a willingness to enter the circuit of organized paths, precisely because of the beautiful human and supportive climate that is created during the shared experience.

"A way to live the territory that we personally live day-by-day vertically and horizontally, moving between past and present, but essentially linked to our agricultural and pastoral culture. We need to know in-depth the territory and its routes for being able to narrate it at the best . . . " (Interview of Marcello Pastorini, 19 January 2021).

In the logic of the three pillars of ecological, socio-cultural, and economic sustainability, the first two are certainly amplified and fully achieved in this case, while it lacks in part the sustainability and guarantee of replicability of the experience in economic terms. The absence of full political recognition and financial support, albeit inaugural for purely managerial purposes, implies the need for a progressive privatization and structuring of the ecomuseum, offered in the form of a cooperative circuit that could thus guarantee the hope of continuity and profitability for the territory and the communities touched by the routes.

The ecomuseum, highly critical in terms of bureaucratic delays and the lack of territorial governance of development processes, places itself in a consciously resilient position with respect to the more structured forms of rural development and tourism directed from above, strongly claiming effectiveness, self-organization, gratuitousness, the spirit of solidarity, and the distance from territorial powers, which represent one of the most interesting keys in the proposal of a conscious and, in a certain way, assertive tourism as an idea for the responsible and sustainable use of the territory and of cultural expressions and assets of local communities.

While being interviewed about criticalities in developing the ecomuseum proposal, Marcello Pastorini commented: "The regional law on Ecomuseum has never received its implementing decrees. There is a structural lack of means and of circularity between institutions and group of associations on the territory. No funds, no political support in order to develop an integral proposal despite funds apparently consecrated to pastoral routes regeneration and recovery. At the same time, we work with local farms and agrifood producers or tourist operators, but no cooperation with the overall tourist system has never been established. All our activities are totally spontaneous and free, included all our

efforts to collect information and local historical and demo-anthropological knowledge in the local dimension" (interview of Marcello Pastorini, 19 January 2021).

5.2.2. The Institutional Contract of Development (CIS) for the "Tourist Development along the Molise Pastoral Routes"

At a political level, we can observe some criticality in legal frameworks for conservation/valorization policies, as well as in the ecological movements' activism or the heritage communities' concerns. Many rules, as we saw, are blurred, as well as regional planning and the systems of the distribution of the public domain to private citizens, even if community vigilance has recently increased.

There has been a flourishing of initiatives, conferences, paths and walks dedicated to the protection, rediscovery, and enhancement of sheep tracks in Molise and beyond: new funds such as, for example, the specific measures of the rural development program, LAGs projects on the territories, national strategy for inner areas actions, and, more recently, the CIS, giving to Molise an important fund for the "Tourism development along the Molise pastoral routes" divided into the "recovery and enhancement of the tourist path" and the "incentive and enhancement of the tourist offer" (with a financial endowment of almost EUR 130 million) reserved for related actions and projects of valorization. The research and data collecting about this complex and long definition process has been realized through the direct participation of some of the authors in brainstorming meetings, preparation of factsheets preliminary to the presentation of the project, feedback meetings with representatives of different local institutions, and other local and superlocal stakeholders from January 2020 until today.

The project, formally led by a small municipality (Campodipietra), and based on a consortium of 59 local communities, was mainly divided into structural interventions in building renovations and territorial restoration along the pathways that, in the case of Molise, represents the most part of the regional territory, since each part of Molise is crossed by royal cattle tracks and small cattle tracks, by arms, sheepfolds (defènze, stazzi), and places where fairs and markets for livestock and products related to sheep farming were held, where wool was processed as well as cheeses and meats.

That the regional leadership has strongly insisted in its public communication on sheep tracks and transhumance is represented today as an opportunity for development and for an overall rethinking of the tourist offer and promotion of artisan products and related agro-food chains. Nonetheless, the process is to be evaluated if real, though coming from a network of municipalities declaring its transversal and participatory vocation could consolidate a strategic perspective, really enabling an integrated framework. Such a project can allow, in our opinion, the opportunity to realize and give substance to a concrete experience of the ecomuseum of transhumance based on the sheep-track network of Molise, profiting from previous experiences in Italy and in wider Europe. We think about an ecomuseum of transhumance as a "moving museum", strongly characterized by walking ways and tourist/cultural offers and suggestions, coping with the increasing demand for rural and experiential tourism, as described and outlined at the very beginning of this paper. This can imply some difficulties in establishing boundaries and info/enter points for the area interested in such a kind of ecomuseum, including the overcoming of a certain vagueness or dispersion, or some problems of conservation regarding material heritage assets. At the same time, we have documented throughout our interviews and ethnographic observations how much ecomuseums are embedded and deeply rooted in the local culture, implying the social commitment of these communities to the value of bio-cultural heritage, perfectly coherent with the European Council's Faro Convention (2005), recently ratified by Italian Government (December 2020).

These kinds of 'heritagizating' processes questions the link between local practice, landscape conservation, and cultural heritage, implying a strict cooperation among disciplines and scientific competences, governance and political visions of the territory. At the same time, we are faced with a powerful and challenging issue: can pathways—religious, cultural, fitness, and wellness pathways—really be considered one of the most important

and innovative tourist perspectives for sustainable and experiential tourism, as well as for territorial development and the heritage communities' empowerment?

## 6. Awareness, Agenda, Action

The two territories and three cases in question are located in different stages of the process of empowerment of rural tourism, both because of the different territorial scales underlying them, and because of the time necessary for it to take place, the duration of which is obviously proportional to the presence of local conflicts of interest and to the ability of the promoters to resolve them.

In Figure 2, the representation of the triple-A model proposed by Mihalic [42] has been reworked into a system of concentric triangles showing how such a kind of empowerment is not always symmetrical. In fact, unlike the Piedmont case, the two Molise cases are based, on the one hand on private initiatives, which do not receive (or prefer to avoid) the necessary support from local and/or regional governments, and, on the other hand, on public initiatives, which prefer or which fail to activate effective forms of governance, also because they are still significantly far from achieving full awareness of the sustainability of tourism and the diversification processes.

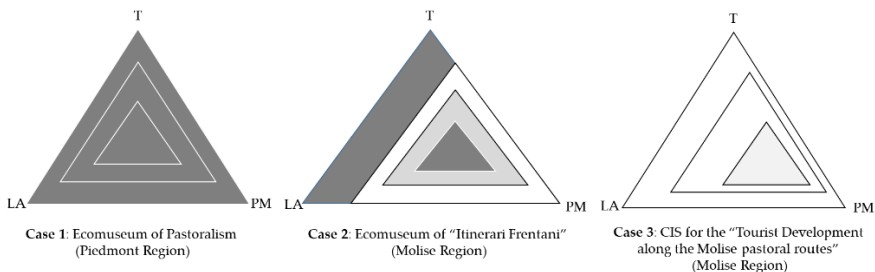

**Figure 2.** Triple-A model referred to the three cases. In each figure the match among different triangles represents the stage of 'responsustanaibility' reached by each case. The internal triangle corresponds to the awareness, the external one to the action. The intensity of the shading expresses the degree of penetration of sustainability in the conduct of the actors represented at the top (LA = Local Actors; PM = Policy/decision-makers; T = Tourists).

As emerged from the fieldwork, the ecomuseum was an effective tool for the transition from awareness to agenda, consisting of a prolonged and articulated strategic program, in which, as noted above, the role played by both regional and intermediate institutions are fundamental, and that, together, they have managed to generate public resources to the project, aimed at promoting sustainable and responsible forms of rural tourism of an experiential type, therefore characterized by a strong cultural value. In this case, we noticed the consistency of a small very cohesive village's engagement, started around a core of productive revitalization and the conservation of farmed biodiversity that immediately gave a strong community imprint, but also a productive one that would later lead, shortly thereafter, into an ecomuseum proposal. The effective synergy between local associations, spontaneous collectives, and a consortium of companies then found the necessary support, in this case, in the local LAG and the Mountain Union, to structure itself into a proposal for territorial enhancement, which was subsequently implemented in a form recognized, regulated, and financed at the regional level by the ecomuseum. At the local scale, such an institutionally recognized structure has also supported and accompanied the development of a capillary activity of the support and advice of citizens in the restructuring and start-up of agritourism activities and widespread rural accommodations that ended up generating a discreet effervescence of tourist activities in the area, despite their geographical marginality.

The context of the Molise region is, somehow, more complex. The strong engagement in the recently successful process of heritagization has engendered and developed at a local level in a different way to deal with this cultural asset and potential tourist resource.

The case of the explicitly self-defined 'Ecomuseum Itinerari Frentaini' (EIF) is, with respect to the analysis model applied, only partially completed, since a still small number of territorial actors participated in it, but above all because it appears to lack the necessary involvement of local, intermediate, and regional institutions capable of transforming the experience of the founders of the same in a true model of territorial regeneration based on experiential tourism. As Mihalic [42] has pointed out, the change of state can only be partial, representing a good practice to be transformed into territorial regeneration strategies in the context of wider public policy frameworks concerning rural and regional development, as it would seem to have happened. In such a case, we noticed on one hand the wide awareness of what has to be considered and developed as an ecomuseum, namely a strong commitment of a spontaneous group of local actors and associations, scarcely considered at institutional scale. In such a case, the local community seems to be gradually becoming aware, accompanied in this by the solicitation and cultural guidance of ecomuseum actions, of value to local bio-cultural heritage as a territorial resource and as a lever for "responsustainable tourism" development. Furthermore, the ecomuseum seems to have brought a fruitful activation of a circuit of friendly companies, made protagonists precisely by the spontaneous and non-institutionalized participation in the activities of the ecomuseum. In the background, there seems to be an availability to the logic of the development, and with it also of tourism, responsibly and sustainably represented by the inaugural planning of a local bio-district which, created as a space for the affirmation of potential local political aspirations, seems to struggle to think about forms of interaction between agricultural production of excellence and responsible landscape and tourist offers (the association between good landscape and good products, for example the famous local oil and the development of small dairy farms along the still traceable and practicable paths).

The case of CIS, explicitly aimed at the "Tourist development along the Molise pastoral routes", would seem to have instead a configuration diametrically opposite to the previous case. In this case, in fact, a conspicuous availability of public resources has been explicitly addressed to the development of tourism along the sheep tracks, but it would seem, unfortunately, absolutely lacking in a systemic and strategic project oriented to the re-stabilization of rural tourism. As has emerged from the figure above, the state of awareness of the ecological and social dimensions of tourism sustainability would seem insufficient to allow an effective transition of state towards planning, thus limiting the action to simple material investments, mainly without a significant link with sheep tracks and pastoral routes and with the inestimable value of the intangible heritage connected to them, on which the concept of sustainable and responsible tourism must be based. Thus, the entire design system seems almost exclusively marked by the material recovery of the paths and built structures through a poor coherence of the project interventions. What is missing, most essentially, seems to be an adequate action aimed at a territorial regeneration ranging from the shared planning of the enhancement of paths to the realization of concerted tourism promotion activities at the level of project coordination. Above all, we observe the absence of projects aiming at the revitalization of productive activities, intended both as an element of sustainable territorial development and as a form of sustainable and responsible tourism experience.

What seems to be missing in such a financed and ambitious project is precisely the transhumance, though continuously evoked in the public discourse, as an activity and as a practice, in the presence of animals and breeding activities, the absence of a network of info-points and territorial interpretation centers and heritage recognition of this area and as a real territorial brand. The awareness of the heritage value here seems not to have translated into a political and planning agenda, although this limit is detected in the local coordination of the consortium.

In this sense, we have accurately considered the positive reception and the attempt to introduce in the rather narrow mallets of the very centralized CIS design of a "plan of intangible actions" proposed by the local university, which essentially aims to compensate

the project for this weakness with cultural and tourist promotion as well as participatory activities of biocultural landscape regeneration, aimed at creating an almost regional ecomuseum itinerary based on sheep tracks network and transhumance practices.

## 7. Conclusions

Throughout our review of the scientific literature on ecomuseums both in the socio-anthropological as well as in rural economy and tourism destination management debate, we have tried to define a model for an ecomuseum of pastoral bio-cultural heritage and its building of tourist destination.

Based on a strong participatory and bottom-up approach, ecomuseums radically imply a strong territorial embeddedness, a sustainable use of places and local human resources, and a valorization of a local knowledge–practice system.

In the case of transhumance, the recent UNESCO nomination and the consequent global visibility and dissemination imposed to reconsider the representation and narrative of the practice and the framework in which a transhumance promotion has been realized.

The comparison between two Italian cases—the EPP and the on-going processes of heritagization and valorization of the horizontal transhumance in the southern-central regions close to the inner area of Italian Apennines has been extremely useful.

Through this comparison and parallel observations, we noticed that the value of a well-articulated and stable regulatory framework (a regional law since 1995, with its concerned implementing decrees) for the development and sustainability of an ecomuseum based on pastoral activities and promoting the territory generally interested by this knowledge–practice system represents a best practice only in the first case, while in the second, after the endorsing engagement of the Molise region, the ecomuseum model has been totally dismissed. This last case provides, in fact, the evidence of the tenacious and somehow heroic continuation of the experience by a few determined promoters of the model, willing to continue even in absence of the regulatory system necessary for the formal recognition of the eco-museum, now considered no longer useful and necessary. This has been the choice, on one hand, to avoid wasting the significant position of income earned through the patient work of weaving dense networks of relationships between rural tourists and local economic operators, and, on the other, the increasing disengagement of regional and local institutions (i.e., the LAGs), undervaluing the economic potential of the model and proposing rural development strategies that are essentially based on poor innovation and incisiveness, while the tourist diversification of agriculture, essentially rooted in sectoral and productivistic logics, remain very fragmented.

The completion of the regulatory process of ecomuseums in Molise would therefore seem a necessary, but not sufficient, condition for experimenting with a new organizational model of local bio-cultural resources, enabling the promotion of original forms of economic and social regeneration of the innermost areas of the region, focusing precisely on the ecomuseum model both in the programming of resources destined for rural development within the new program-cycle of the CAP, and in the implementation of the CIS project specifically intended for the tourist enhancement of sheep tracks.

A very interesting insight of this research, then, is the definition of a set of elements characterizing the actions and aims of ecomuseums in Italy: community maps elaboration as a form of community empowerment; participated inventories of the knowledge–practices system as a way to give substance to the territorial narrative; private/public cooperation in commercial/tourist exploitation and impact of pastoral and other collateral activities on the community and on the territory; connection with the policies engendered by the leader strategy in the local context, as the LAG presence, though marginal to the very center of the ecomuseum, offers expertise and a counselling system to farmers and herders, and so on.

Through the three different cases, the research has isolated positive and negative elements of transhumance and sheep tracks as a driver of a sustainable and responsible tourist offer:

1.  In EPP, the positive influence of a more strict and efficient multi-level cooperation among local institutions, LAGs, Mountain Union, different scale strategies and programs, and diffused and informal local groups;

2.  In CIS, the persistence of a top-down in valorization deriving from the global UNESCO recognition and the consequent institutional actions of promotion of this bio-cultural heritage resource;

3.  In EIF, the resistance of more or less informal local groups in copying this new phase and moment of the conservation and valorization process, despite the opportunity for them to be sustained relying on their capillary action to raise the awareness of the local population about the value of what they share.

Starting from a powerful contamination among disciplines—bio-cultural heritage and environmental studies, rural economy, geography, as well as biodiversity and agrarian studies—pastoralism has allowed us to discuss, through these specific cases, environmental sustainability, ecological approaches and the participation in decision-making processes, and, for the governance of the territories, a strategy for inland and peripheral areas for the fighting of the depopulation and marginality of global processes and biodiversity and cultural diversity reduction, according to the mainstream major global agencies concerned.

The resilience of such an ancient practice, is then entangled with an heritagization process, transforming pastoral practices and routes in a potentially global tourist attraction, being transhumance at an almost global phenomenon and being now much more globalized due to the recent UNESCO nomination.

In EPP, as in EIF, such a sociocultural dimension, as well as the community commitment, seem to be structural and essential to the model; in the CIS project, we assisted in a top-down process of generation, often evoking the need for collective participation, but, in reality, excluding local populations by the possibility of contributing and concretely fertilizing the genesis of the shared local action.

In summary, it is possible to add that all of the cases and their respective analyses presented leads to a fundamental final consideration: without a strong community awareness and a fully shared and participatory agenda, it is impossible to realize the desirable synergy between responsibility and sustainability, enabling the revitalization of the local productive practices and non-episodic actions of the cultural and tourist-related valorization of local territories.

**Author Contributions:** Conceptualization, L.B., A.B. and C.I.; Methodology, L.B. and A.B.; Investigation, L.B.; Data curation, L.B.; Writing—original draft preparation, L.B.; Writing—review and editing, A.B., L.B. and C.I.; Supervision, C.I.; Project administration, A.B. and L.B.; Funding acquisition, C.I. All authors have read and agreed to the published version of the manuscript.

**Funding:** The APC was funded by DESCLAM project of Department of Agricultural, Environmental and Food Sciences of University of Molise.

**Institutional Review Board Statement:** Not applicable.

**Informed Consent Statement:** Informed consent was obtained from all subjects involved in the study.

**Data Availability Statement:** All the interviews are available in a common author's drive which eventually can be shared on demand.

**Conflicts of Interest:** The authors declare no conflict of interest.

## Appendix A

Ecomuseo della pastorizia, Frazione Pietraporzio, Comune di Pontebernardo, CN—http://www.visitstura.it/cultura-e-arte/attrattive/musei/pietraporzio-ecomuseo-della-pastorizia/ ((accessed on 29 June 2021)). The ecomuseum of pastoralism is located in the small town of the province of Cuneo, in the region of Piedmont, Pontebernardo (80 inhabitants), which is a fraction of the municipality of Pietraporzio in the province of Cuneo of the Italian Region of Piedmont. It is a mountain village (h. 1246 extension), located in the Val Stura, in an

area historically characterized by the transhumance routes between the Italian mountains and the French lowlands in which the herds remained grazing in the rigorous winters of the valley. Piedmont is in effect the first region to make a regional law on ecomuseums in Italy in 1995 and even today is one of the most advanced in the development of a new model of open-air museums and in animating strong community participation and commitment to it. The ecomuseums were officially inaugurated in 2000, but it has been implicitly active since the mid-eighties, when the recuperation of Sambucana sheep and the concerned consortium was created. The headquarters of the ecomuseum is located in a building in the center of the city that, acquired by Mountain Union, has been renovated to be available for community activities. On the ground floor there is a small dairy that allows the families of shepherds living in the area to prepare an excellent sheep cheese, the "Toumo dell'Ecomuseo", and a laboratory for processing the meat of the Sambucana sheep, with which they produce excellent sausages. In the same building, there is also the tasting point of the ecomuseum, inaugurated in 2008. A small square divides this first structure from a second building that houses on the ground floor the Arieti Center, managed by the L'Escaroun consortium, while on the top floor is the interpretation center of the "Na Draio per Vioure", an Occitan sentence meaning "a walkways for living". The context in which this ecomuseum has developed is the area of the valleys of the Cuneo at the end of the last century. In this period, the area was one of the most marked by depopulation and the rapid deterioration of most of its traditional production activities: mainly sheep farming and related activities, such as the manufacturing of cheese, crafts, and sausage-making. The incidence of tourism was decidedly limited, compared to the urban poles of the rest of the region (Turin, in particular, but not only) or other areas related to wine tourism (for example, the Barolo area and the entire area of Langhe as a whole).

Regarding the cooperation of the EPP with the French Maison de la Transhumance, we can consider that the Maison de la Transhumance contains aspects that bring it closer or characterize it as a wider container of territorial safeguarding and promotional/regenerative activities related to sheep farming similar, but not identical, to the EPP, because of its larger territory and missions. The Maison, moreover, articulates, coordinates, and functions as a collector of experiences, resources, and projects destined to generate new local opportunities by effectively putting them on the net with each other.

**Appendix B**

It is the case of the Colantuono from Frosolone, a family of traditional herders that continue to come from the mountain of the Molise to the plane of the Puglia with their cattle every year. Meanwhile, other families of shepherds and herders are revitalizing small–medium scale transhumance in connection with contextual academic research. Antonio Innamorato, for example, has recuperated the ancient transhumant track from Campitello Matese to Sepino, an archaeological area of Molise inhabited and built since the Samnites' period and successively implemented and enlarged during the Roman Empire. The settlement is one of the most important and well conserved of this area and it was evidently connected to transhumance as a sheep market and fence, as well as a cheese/wool transformation activities center [78].

Concerning the Colantuono, they started the transhumant revival since 2008, supported by the Local Group of Action (LAG), proposing again the slow move of the cattle along the traditional 'green highways' from Frosolone in Molise to San Marco in Lamis farm in Puglia, at a distance of 150 km, as their ancestors already did (interview of Carmelina Colantuono, 12 November 2020). During the last decade, this revitalized transhumance has become a real tourist event to which many local associations, public institutions, and private citizens aim at participating in. In the same years the Local Group of Action has been involved and promoted transhumant culture to the UNESCO Intangible Cultural Heritage List: a process ended in 2019 with the inscription of transhumance as an international network issue to the ICH UNESCO List with Greece and Austria. At a closer observation, the overall process reflects a rather top-down organization and a moderate commercial-oriented trend. The dossier, realized thanks to the cooperation of a group of

experts coordinated by the Ministry of Agricultural, Food and Forestry Policies, though considered exemplary in terms of consistency, has been only partially discussed and shared with local communities, and it is not so fully representative of local and different instances.

Concerning the Innamorato case, a 2-day event during September of 2017 summarizes, somewhat, the work done by this family with experts on the bio-cultural heritage of transhumance in order to recover the memory of the ancient local narrations and to return it to the locals (interview of Antonio Innamorato on 23 November 2020).

**Appendix C**

Marcello Pastorini, who is the principal animator of the association and of the ecomuseum approach, has collected in over a decade, a thick knowledge of ceremonials and rituals, popular literature, oral poetry, traditional songs and material culture practices, and has experienced all these records and witnessing of local culture in a very performative way, along walking ways organised in a very autonomous and non-structured way, using word of mouth and the informal network of knowledge already acquired, of those who have already had the experience of walking with the promoting group of the ecomuseum and of the "Cantori della memoria", who are the performative 'soul' of this group.

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
