# Peer review of "Walking along the Sheeptrack…Rural Tourism, Ecomuseums, and Bio-Cultural Heritage"

_sustainability, doi:10.3390/su13168870_

Round 1

Reviewer 1 Report

Walking Along the Sheeptrack…Rural tourism, ecomuseums, and bio-cultural heritage
What is the main question addressed by the research?
• The main question addressed by the research is the development of sustainable regional tourism in terms of tourism potential in pastoralism and all the factors that can contribute to the formation of the sustainable tourism product (local food, rural diversification, tradition, cultural heritage).
Is it relevant and interesting?
• The subject is relevant for innovation and sustainable regional development, the case of ecomuseum of pastoralism is very interesting as tourism product, the topic of this research must be studied in other countries.
How original is the approach?
• Ecomuseums of pastoralism and transhumance as potential drivers for development and territorial regeneration turned in tourism potential it is an original approach.
What does it add to existing publications on related topics?
• The paper presents a model for sustainable regional tourism development in the context of the existence of an ecomuseum and developing a sustainable tourism potential.
Is the paper well written?
• The paper is well written, maybe too long, should be reviewed following the suggestions.
Are the conclusions consistent with the evidence and arguments presented?
• The chapter Conclusion do not exist, but some comments are relevant from the rows 807-819. I consider important and necessary the conclusion as a separate part of the paper. This part must be added. I mentioned this in review.

Please note and correct - 

*Acronyms/Abbreviations/Initialisms should be defined the first time they appear in each sections

WTO = World Trade Organization.

UNWTO = World Tourism Organization

row 313 - " a specific" appears twice

row 39 - use the pagination for the embedded text

* I consider the conclusions section necessary 

Author Response

Dear Referee,

Thank you for your useful comments and suggestions on our manuscript and for the opportunity to improve it.

Please, you can find enclosed a revision of our manuscript entitled «Walking Along the Sheeptrack…Rural tourism, ecomuseums, and bio-cultural heritage».

We have considered all your suggestions and incorporated them into the revised version.

Changes to the original manuscript are highlighted both using the "Track Changes".

Comment 1

*Acronyms/Abbreviations/Initialisms should be defined the first time they appear in each sections

Acronyms have been defined the first time they appear

For practical reasons and agility of lecture we provided also acronyms for the three case-studies we outlined. The acronyms have been solved and explicated the first time the name of eache case-study appears

Comment 2

row 313 - " a specific" appears twice

this repeat was removed

Comment 3

row 39 - use the pagination for the embedded text

done

Comment 4

* I consider the conclusions section necessary 

Conclusions have been explicitly added

Reviewer 2 Report

This manuscript reflects an interesting topic and is related to an emerging research field in the context of rural tourism. The authors have put a lot of effort in providing a thorough analysis of the importance of ecomuseums with respect to the sheeptracks. With that said, I do have several suggestions for improving the manuscript prior to potential publication. 

First of all, the Introduction must be revised. It is too long narrative and in its present form it consists of what can be considered a theoretical background. These two parts must be separated, but in a way that theoretical background is logically divided into subheadings for the purpose of clarity of the analysis. 

Furthermore, even though it is clear that the authors generally point out and emphasise the balance between three pillars of sustainability, there is one sentence where is seems as if the socio-cultural criteria is of minor importance (rows 236-237). This should be elaborated in more details so that it is clear what is meant by that statement.  

In the second section, objectives must be clearly emphasized and in concluding remarks the authors should reflect on those objectives so that it would be more clear if they were met and to which extent. Also, this section should be more precise and not so descriptive. 

In the third part, the authors mention applied research methodologies which are to be used in the manuscript. However, it is not clear how the data were collected, how and when ad-hoc interviews were conducted and which data is a result of empirical research. This should be clearly explained in the fourth section. 

The fourth section is too narrative and occasionally too descriptive. This leads to the issue of not referring to source of data when providing characteristics of certain tourist segments (e.g. rows 460-465). Due to the extensive explanations it is somewhat difficult to recognize precise connections between theoretical findings and authors' own research. I suggest that this part is revised in a way that only key findings are left in the main body of the text, while descriptive parts are relocated into appendixes. 

Throughout the fourth section it is necessary to refer to data sources or to emphasize if the analysis is a result of primary research.

Finally, I suggest that limitations of the research and suggestions for future research are emphasized. The scientific contribution of the research should be more clearly pointed out.

I believe that the authors will manage to revise the manuscript according to these suggestions, as their work is related to extremely important issue of preserving rural areas by embracing sustainable policies. 

Author Response

Dear Referee,

Thank you for your useful comments and suggestions on our manuscript and for the opportunity to improve it.

Please, you can find enclosed a revision of our manuscript entitled «Walking Along the Sheeptrack…Rural tourism, ecomuseums, and bio-cultural heritage».

We have considered all your suggestions and incorporated them into the revised version.

Changes to the original manuscript are highlighted both using the "Track Changes".

Comment 1:

First of all, the Introduction must be revised. It is too long narrative and in its present form it consists of what can be considered a theoretical background. These two parts must be separated, but in a way that theoretical background is logically divided into subheadings for the purpose of clarity of the analysis. 

Introduction has been revised and divided in two parts: “Introduction” and Ch. 2: ‘theoretical background’ divided in two paragraphs: “2.1 Motivations of tourist and rural touristic models” and “2.2 Rural tourism as a creative sustainable and responsible experience of leisure”.

Comment 2:

Furthermore, even though it is clear that the authors generally point out and emphasise the balance between three pillars of sustainability, there is one sentence where is seems as if the socio-cultural criteria is of minor importance (rows 236-237). This should be elaborated in more details so that it is clear what is meant by that statement.  

This specific sentence has been solved and clarified (now 236-247).

Comment 3:

In the second section, objectives must be clearly emphasized and in concluding remarks the authors should reflect on those objectives so that it would be more clear if they were met and to which extent. Also, this section should be more precise and not so descriptive. 

In the 2nd Section (2) we made major changes: 1) reorganized a bit the topics; 2) outlined more precisely the theoretical framework, splitting in two sub-sections and 3) adding, at the very end of the paragraph, what we consider a more clearly defined working hypothesis.

Comment 4:

In the third part (this became: 4th section), the authors mention applied research methodologies which are to be used in the manuscript. However, it is not clear how the data were collected, how and when ad-hoc interviews were conducted and which data is a result of empirical research. This should be clearly explained in the fourth section. 

In the 3rd Section (3), particularly, we have tried to clarify and specify some points, solving the more difficult sentences.

In the 4th Section (4) we added a description of all “Methods and Materials” (Ch. 4) on methodologies and materials used, read, commented for the background research, all the elements considered for a socio-economic evaluation of case-studies and all the and ethnographic materials collected during the research: cartographies of the two regions/three cases observed, brief outline of them, precise definition of specific methods employed for collecting local data and testimonials.

Comment 5:

The fourth section (this became: 5th section) is too narrative and occasionally too descriptive. This leads to the issue of not referring to source of data when providing characteristics of certain tourist segments (e.g. rows 460-465). Due to the extensive explanations it is somewhat difficult to recognize precise connections between theoretical findings and authors' own research. I suggest that this part is revised in a way that only key findings are left in the main body of the text, while descriptive parts are relocated into appendixes. 

About the 5th section (5. Two territories, three cases) of the paper, in order to made easier and clearer such a section as well as the understanding of the main data and objectives of present research, we have moved to several Appendixes the parts we considered more descriptive, though useful and needed to the overall comprehension of our concept and interpretation. So, we have added:

Appendix A (already present in the previous version): we add some more materials from the body of the text.

Appendix B: An Appendix for the very general reference to the phenomenon of transhumance in Molise.

Appendix C: An Appendix to outline more in detail the “Ecomuseum Itinerari Frentani” genesis and concept.

Comment 6:

Throughout the fourth section (this became: 5th section) it is necessary to refer to data sources or to emphasize if the analysis is a result of primary research.

Throughout the 5th section: Precise references to data collection process (methodological issues and specific sources, contextual observation, semi-structured and in-depth interviews, consulted documents) have been added as requested.

Comment 7:

Finally, I suggest that limitations of the research and suggestions for future research are emphasized. The scientific contribution of the research should be more clearly pointed out.

The 7th section (Conclusions): have been slightly extended to define more clearly the results of the research and moreover to outline the effectiveness of the application of the chosen Triple-A model (Awareness, Agenda, Action) as well as the “responsustainable tourism” interpretative key, applied since the beginning to verify the starting working hypothesis of the paper.

Round 2

Reviewer 2 Report

The authors have put a lot of effort in revising the manuscript and the quality of the paper is now improved.